# Work-related stress and sleep problems among small-scale miners in Ghana: The role of psychological factors

Emmanuel Nyaaba[1]*, Vanessa Francesca Epis[1], Lawrence Guodaar[1], Razak M. Gyasi[2,3]

1 Department of Geography and Rural Development, Kwame Nkrumah University of Science and Technology, Kumasi, Ghana, 2 African Population and Health Research Center, Nairobi, Kenya, 3 National Centre for Naturopathic Medicine, Faculty of Health, Southern Cross University, Lismore, New South Wales, Australia

* nknyaaba@gmail.com

## Abstract

### Objectives

Sleep problems (SP) are prevalent among small-scale miners, yet little is known about their psychological and occupational determinants. We examined the association between work-related stress (WRS) and SP among small-scale miners in Ghana and explored the mediating roles of anxiety and depression in this association.,

### Methods

In this community-based cross-sectional study, data were collected from 664 miners in Obuasi,Ghana. WRS and SP were assessed using the Perceived Stress Scale (PSS-4) and the WHO Disability Assessment Schedule 2.0 (WHODAS 2.0), respectively, while anxiety and depression were measured using the GAD-7 and PHQ-9. Adjusted multivariate regressions and bootstrapped serial mediation models (Hayes' PROCESS Model 6) evaluated the hypothesized pathways.

### Results

The mean (SD) age was 28.8 (8.2) years, and 84.3% were males. WRS was significantly associated with SP (B = 0.2964, 95% CI = 0.2398–0.3530), with both direct (42.5%) and indirect (57.5%) effects. Anxiety mediated 38.0% of the total effect, depression 11.4%, and the anxiety–depression pathway 8.1%.

### Conclusions

The study findings suggest that anxiety and depression serially mediate the WRS–SP link. These results underscore the need to address occupational stressors and incorporate mental health support into workplace policies to improve sleep quality and overall well-being in this vulnerable workforce.

**Data availability statement:** All relevant data are within the manuscript and its Supporting information files.

**Funding:** The author(s) received no specific funding for this work.

**Competing interests:** The authors declare that they have no known competing financial interests or personal relationships that could have appeared to influence the work reported in this paper.

## Introduction

Sleep quality is a key clinical and epidemiological measure linked to various health outcomes. Poor sleep has been associated with physical conditions such as obesity, metabolic syndrome, and cardiovascular disease [1,2], as well as psychological issues like cognitive decline, low mood, aggression, and depression [3–5]. Sleep problems (SP), including difficulty initiating or maintaining sleep, bad dreams, and sleep medication use, are common across occupational groups, notably among miners [6–8].

While growing evidence highlights occupational stressors as key predictors of poor sleep outcomes, one important but often underexplored factor is work-related stress (WRS). Defined by [9] as emotional, cognitive, behavioral, and physiological responses to negative aspects of work, WRS reflects a state of heightened arousal and distress. It often arises from persistent job demands such as excessive workload, role ambiguity, low job control, and employment insecurity [9]. Although short-term stress responses, such as fatigue, elevated heart rate, and negative affect, are common and typically harmless, they do not disrupt sleep when the body adequately recovers during off-job periods. However, sleep may be impaired when stress recovery is insufficient and physiological systems, such as the hypothalamic-pituitary-adrenocortical and sympathetic-adrenal-medullary axes, remain activated during rest periods [10]. This prolonged activation contributes to cumulative biological strain, which can ultimately result in serious health conditions [11]. WRS, including high-strain, low-control, or effort-reward imbalance situations without adequate recovery has been shown to predict poor sleep outcomes across various populations [12,13]. For instance, a mid-life Australian cohort study found that job stress, assessed via the Job Content Questionnaire (JCQ), independently predicted chronic sleep disturbances [12], while a systematic review of 38 studies confirmed a consistent negative association between WRS and sleep quality [14]. Among Chinese oilfield workers, occupational stress, which was assessed using the Occupational Stress Inventory-Revised (OSI-R) scale, capturing role overload, role ambiguity, and responsibility was positively associated with increased sleep [15].

Despite these crucial findings from Western countries and some Asian societies, data on the association between WRS and SP in lower-middle-income countries (LMICs), particularly in the Ghanaian setting, is scarce. Crucially, while research has established clear links between occupational stress and sleep quality in formal work settings, the unique challenges faced by informal workers, especially in LMICs, remain understudied. This knowledge gap is crucial, given that LMICs often present unique work-related stressors, including informal employment arrangements, limited labor protections, low wages, high job insecurity, limited access to psychosocial support, and under-resourced health systems [6,20,21], all of which may exacerbate the effects of WRS on physical and mental health. These contextual differences therefore suggest that findings from high-income countries may not fully capture the lived realities of workers in LMICs, where structural vulnerabilities heighten the risk of cumulative stress and poor sleep outcomes.

Importantly, the adverse effects of WRS on SP may also operate through psychological pathways. Evidence suggests that WRS contributes to increased anxiety and depression, which can disrupt emotional regulation and sleep patterns [16,17]. Anxiety, often triggered by chronic stress, can elevate arousal levels, making it difficult to initiate or maintain sleep, while depression exacerbate these issues by influencing sleep architecture and cognitive processes related to rest [18]. Anxiety and depression frequently co-occur, with anxiety often preceding depression [18]. While these mechanisms are well-documented in high-income countries, studies from LMICs remain limited. For instance, a study of 2,889 Chinese nurses found a significant association between occupational stress and poor sleep, mediated by anxiety and depression [19]. However, research examining the WRS–SP link in LMICs, particularly among small-scale miners, is scarce. Understanding these psychological pathways is crucial for designing targeted interventions to alleviate SP and improve mental health among vulnerable workers in LMICs.

The case for examining these relationships among small-scale miners in Ghana is particularly compelling. Ghana's small-scale mining sector employs over one million people and faces unique occupational challenges, including irregular working hours, physical demands, and exposure to hazardous conditions [20,21]. Despite ongoing policy initiatives to formalize and regulate the sector, limited attention has been paid to these workers' occupational health and safety, particularly their mental health and sleep quality. The informal nature of their work, combined with economic pressures and regulatory uncertainties, creates a perfect storm for occupational stress. Furthermore, Ghana's mental health infrastructure, especially in rural mining communities, remains underdeveloped, making it crucial to understand the pathways through which WRS affects health outcomes. This knowledge gap is particularly significant given Ghana's current focus on reforming the small-scale mining sector and improving occupational health standards.

In this study, the authors aim to examine the association between WRS and SP among small-scale miners in Ghana. The authors hypothesize that WRS will be linked to higher odds of SP and that this relationship will be serially mediated by anxiety and depression (Fig 1).

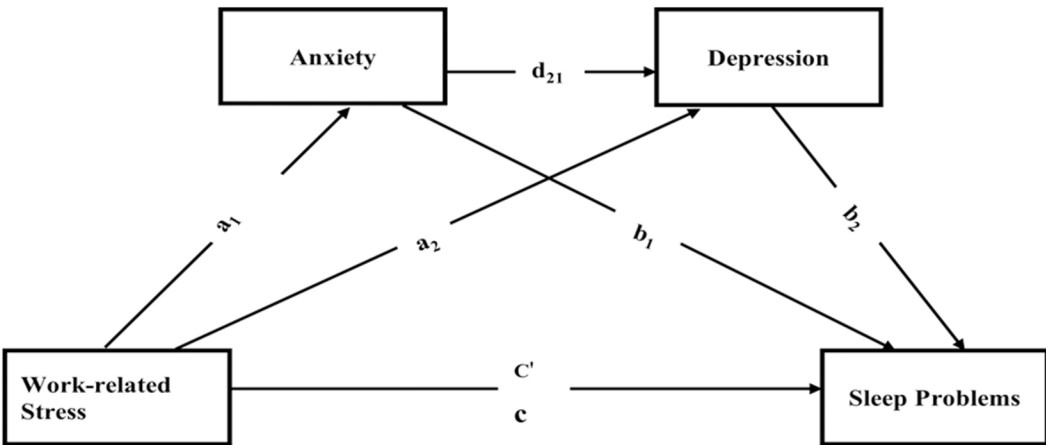

**Fig 1. A hypothesized chain mediation model of the effect of work-related stress on sleep problems via anxiety and depression as potential mediating variables.** Paths $a_1$, $a_2$, $b_1$, $b_2$, $d_{21}$, C', and c, denote path coefficients. Hypothesis 1: work related stress positively affect sleep problems (total effect, c). Hypothesis 2: There is a significant specific indirect effect of work-related stress on sleep problems via anxiety (indirect effect, $a_1$, $b_1$). Hypothesis 3: work related stress have a significant, specific indirect effect on sleep problems via depression (indirect effect $a_2$, $b_2$). Hypothesis 4: work related stress have a significant cascading or serial mediation effect on sleep problems via anxiety and depression (serial indirect effect $a_1$, $d_{21}$, $b_2$).

## Methods

### Ethics

This research adhered to the ethical principles outlined in the Declaration of Helsinki and received approval from the Committee on Human Research, Publications and Ethics (CHRPE) of the School of Medical Sciences at Kwame Nkrumah University of Science and Technology, Kumasi, Ghana (Reference: CHRPE/AP/0103/25). Informed consent was obtained from all participants prior to their participation in the study. Community-level consent was formally documented through signed or witnessed forms detailing the study's purpose, procedures, and voluntary nature, with a lead figure (typically an assembly member) signing on behalf of the community. At the individual level, written informed consent was obtained from all participants. Verbal consent was accepted in cases of literacy challenges, with documentation witnessed and signed. No participants under 16 were included, and no incentives were provided. Participants were informed that their participation was voluntary, and they could choose to withdraw at any time without consequence.

### The survey and participants

This community-based cross-sectional study investigated the association between WRS and SP, with a focus on the potential serial mediating roles of anxiety and depression. WRS was conceptualized as the psychological strain arising from occupational demands including high workload, low autonomy, and limited decision-making power at work, while SP included issues such as difficulty falling or staying asleep and non-restorative sleep. Anxiety and depression were examined as potential psychological mechanisms through which WRS may affect sleep. This framework guided both data collection and analysis. This study was conducted in Obuasi Municipality in the Ashanti Region of Ghana, a region widely recognized for its intensive gold mining activities [21]. The Municipality comprises approximately 32 communities, most of which are noted as mining communities [22]. Out of these, seven communities—Binsere, Nhyiaeso, Gausu, Memrewa, Ntonsua, Apitisu, and Sanso, were purposively selected for this study. These communities were chosen due to their proximity to active ASM sites, the intensity of mining operations, and the prevalence of mining-related health and environmental concerns, making them ideal locations for assessing the impact of work-related stress among miners. Prior to data collection, a reconnaissance survey was conducted in the selected communities to identify and enumerate households with active miners. Working with local residents and district authorities, a detailed household listing was created, focusing on residences with at least one individual engaged in small-scale mining. This survey mapped a total of 1,970 houses, each assigned a unique identification code to construct the sampling frame, warranting the feasibility of a systematic random sampling approach. This technique ensured equal probability of inclusion, minimized sampling bias, and allowed for spatially balanced recruitment across varying community sizes and housing densities.

To ensure adequate statistical power, the study determined the required sample size using the World Health Organization's (WHO) estimation formula [23], considering a 95% confidence interval (CI), a 5% margin of error, and a Type II error (β) of 15%, ensuring 85% power. Given the absence of a reliable population-level prevalence of WRS or SP among small-scale miners in Ghana, we adopted a default prevalence ($\pi$\pi) of 0.5, as recommended when prevalence is unknown. This yielded a minimum required sample size of 481 participants. To account for potential nonresponse, attrition, and missing data, an oversampling rate of 35% was applied, producing a target sample of 649 participants. However, due to high participation rates and field accessibility, the final analytic sample included 664 participants. This sample size meets the theoretical and empirical benchmarks for adequate statistical power (85%), an effect size of 0.25 [24], and robust mediation analysis [25,26]. A flow chart of the selection of the study participants is shown in Fig 2.

$$n = \rho \times \frac{[(\mathrm{U}\alpha_{/2})^2 \times \pi(1 - \pi)]}{\delta^2}$$

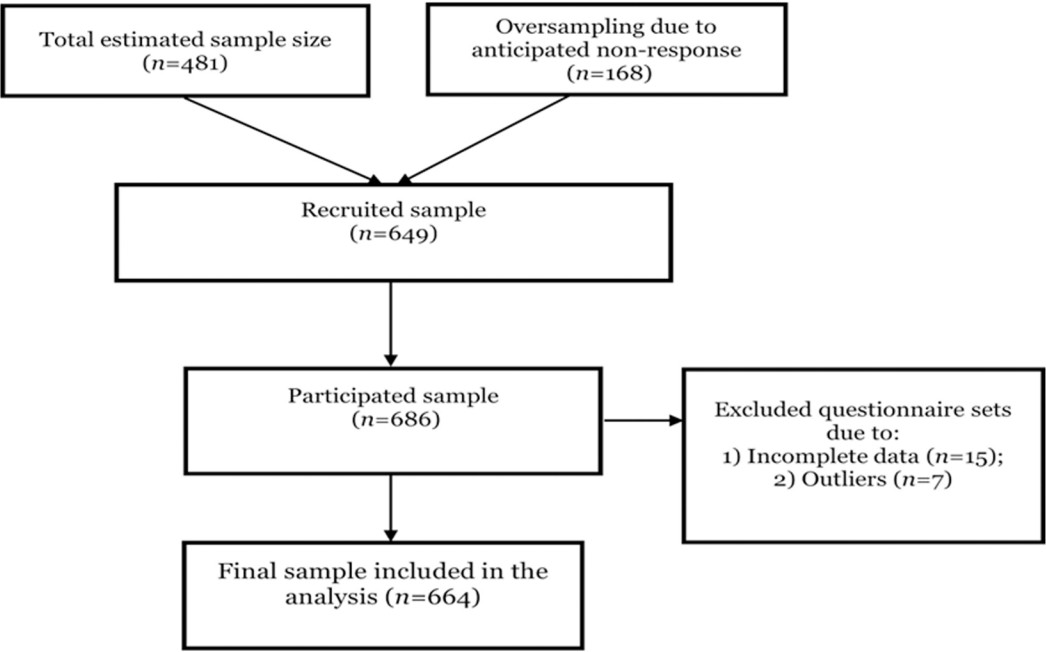

**Fig 2. Flow chart of the selection of study participants.**

This study employed a ʊsystematic simple random sampling technique to select participants both at the household and mining-site levels. First, seven communities were purposively selected for inclusion in the study, as described above. Following this, houses within each selected community were sampled using a spatially guided systematic random sampling method. To ensure balance sampling across the different community sizes and housing density, vary intervals were used to select participants at the community level. In smaller communities (with fewer than 250 houses), every third house was selected, whereas in larger communities (250 or more houses), every fifth house was chosen. In houses with multiple eligible participants one participant was selected using a simple random technique (balloting) to reduce intra-household bias and ensure equal selection probability. If selected houses were vacant, refused participation, or were unavailable after three attempts, replacements were drawn using the same procedure. Additionally, miners were also recruited directly at active mining sites. At each site, eligible miners who were not reached at the household level were approached, and one individual was selected using a simple random sampling technique (balloting). The inclusion criteria required participants to be active small-scale miners aged 18 years or older, with at least six months of continuous mining experience. Individuals who did not meet these criteria or who declined to provide informed consent were excluded from the study.

To enhance efficiency and ensure timely completion, the study employed Rapid Rural Appraisal (RRA) techniques, a participatory research method designed to facilitate data collection in decentralized settings [27]. The research team was divided into four groups, each consisting of five members, ensuring a systematic and coordinated approach to reaching participants across different locations. This division allowed for effective data collection while ensuring that a larger sample size was reached within the available timeframe. This two-pronged strategy ensured that both full-time miners and those who might have been missed through household-based recruitment were included in the study. All measurement tools used in this study (PSS-4, PHQ-9, GAD-7, and WHODAS 2.0) have been previously applied in studies conducted in Ghana [5] or among similar populations in LMICs [16,17], showing acceptable psychometric properties. Where necessary, minor contextual adaptations were made to enhance cultural appropriateness. Instruments were administered in English or translated into Twi using forward and backward translation procedures by bilingual experts. This ensured conceptual

and linguistic equivalence. Additionally, the tools were pre-tested among a small group of 25 artisanal miners to ensure clarity, relevance, and cultural sensitivity before full-scale data collection. Data collection was carried out over 8 weeks (from 16th February 2025–14th April 2025).

To maintain data accuracy and reduce response bias, field teams received comprehensive training in ethical research practices, culturally appropriate engagement, and effective interviewing techniques before data collection commenced. Before fieldwork, the research team formally engaged Obuasi Municipal authorities, who granted permission and helped identify legitimate community representatives (such as township elders, site operators, and assembly members) who acted as custodians of communal consent.

## Measures

### Work-related stress:

Work-related stress was assessed using the 4-item Perceived Stress Scale (PSS-4) [28]. The PSS-4 measures perceived stress on a 5-point scale (0 = never to 4 = very often). Questions assess perceived control and ability to handle difficulties (e.g., "In the last month, how often have you felt that you were unable to control the important things in your life?"). In the current study, the variable showed acceptable internal consistency ($\alpha = 0.72$).

### Sleep problems:

SP were measured using two items adapted from the WHO Disability Assessment Schedule 2.0 (WHODAS 2.0) [29]. Participants were asked; "Overall, in the last 30 days, 1) how much of a problem did you have with sleeping (*e.g.,* falling asleep, waking up frequently during the night, or waking up too early in the morning)? and 2) how much of a problem did you have due to not feeling rested and refreshed during the day (for example, feeling tired, not having energy)?". Each item had five-point options from *none* = 1 to *extreme* = 5. Previous studies have used these items. A total score ranged from 2 to 10, with increasing scores indicating poor sleep quality (Cronbach's alpha = .78 in this study).

### Anxiety:

Anxiety was assessed using the 7-item Generalized Anxiety Disorder (GAD-7) scale [30], which has been widely used to measure anxiety symptoms in older adults. Participants were asked to recall a series of feelings from the past two weeks using seven items with four options each: 0 = never, 1=for several days, 2 = more than half of the days, and 3 = almost every day (scores range: 0–21) with higher scores suggesting more severe anxiety symptoms. The internal consistency reliability of the scale was .79 in this study

### Depression:

Depression was the second mediator assessed using the 9-item Patient Health Questionnaire (PHQ-9) [31]. The participants were asked to recall experiences from the past two weeks, and each item had four options: 0 = rarely, 1=some days (1–2 days), 2 = occasionally (3–4 days), and 3 = most of the time (5–7 days). The total score ranged 0–30; higher score reflected more depressive symptoms. The Cronbach alpha was .80 in the current study.

### *Covariates*:

Several demographics, health status, and occupational variables were included as control variables based on their potential influence on the study variables based on previous studies. Age was measured as a continuous variable in years. Educational level was coded on a 4-point scale (1 = no formal education, 2 = basic education, 3 = secondary education, 4 = tertiary education). Health status was assessed through chronic health conditions, including hypertension, diabetes, respiratory problems, and musculoskeletal problems, each coded dichotomously (0 = absent, 1 = present). Living

arrangements were included as a proxy measure for social support, coded on a 3-point scale (1 = living alone, 2 = residing with family, 3 = living with friends/colleagues). Self-rated health (SRH) was measured using five items from the Medical Outcomes Study Short Form-36 (MOS SF-36). This included a general health rating (1 = excellent to 5 = poor), a health comparison to two years ago (1 = much better now to 5 = much worse now), and three additional items assessing perceived health status (1 = true to 5 = definitely false). A mean score was calculated, with higher scores indicating poorer self-rated health.

*Statistical analyses:*

Statistical analysis was performed using SPSS IBM version 25. Descriptive statistics were calculated for sample characteristics, with means and standard deviations for continuous variables and frequencies and percentages for categorical variables. Zero-order correlations (Pearson's r) were conducted to examine bivariate associations between WRS, SP, anxiety, and depression. The serial mediation models were assessed using the PROCESS macro [32] with WRS as the independent variable, anxiety and depression as mediators, and SP as the outcome. Bootstrapping with 5,000 samples was used to estimate indirect effects and confidence intervals (CIs). Models were adjusted for age, sex, marital status, education, self-rated health, alcohol intake, and chronic conditions. All analyses were conducted using complete case analysis (listwise deletion), as missing data were minimal (<5%) and determined to be missing completely at random (MCAR) based on Little's MCAR test. Statistical significance was determined if the 95% CI for the indirect effect did not overlap zero, with a threshold set at p < 0.05. All data generated and analyzed during this study are included in the supplementary file (S1 File).

## Results

### Sample characteristics

The study participants (see Table 1) had a mean age of 28.82 years (SD = 8.17) and were predominantly male (84.3%) of the sample. The majority of the miners were single (65.1%), followed by married (24.1%), while smaller proportions were cohabiting (7.2%) or divorced/separated (3.6%). Regarding education, most miners had basic education (41.0%) or secondary education (34.9%), while 19.3% had no formal education. The sample was fairly evenly split between native residents (48.2%) and non-natives (51.8%). In terms of living arrangements, about half (50.6%) lived alone, while 39.8% lived with family, and 9.6% lived with friends or colleagues. The average work duration in small-scale mining was 5.95 years (SD = 5.30). More than half of the miners (56.6%) reported alcohol intake. Regarding self-rated health, the most significant proportion rated their health as good (30.1%), followed by fair (26.5%), while excellent (15.7%), very good (14.5%), and poor (13.3%) ratings were less common.

### Correlations

The zero-order Pearson's correlation matrix revealed significant positive associations between WRS and all the main study variables. Specifically, WRS was positively linked to SP (r = 0.371, p < 0.001), anxiety (r = 0.499, p < 0.001), and depression (r = 0.522, p < 0.001), as shown in Table 2. In addition, SP showed a positive correlation with both anxiety (r = 0.442, p < 0.001) and depression (r = 0.387, p < 0.001). Notably, anxiety was strongly and positively correlated with depression (r = 0.586, p < 0.001). These significant interrelations among WRS, SP, anxiety, and depression satisfied the statistical prerequisites for conducting mediation analysis.

### Serial mediation analysis

Fig 3 displays the unstandardized regression coefficients for each proposed pathway. All the hypothesized models demonstrated statistical significance, even after accounting for various potential covariates through robust adjustments.

**Table 1. Sample characteristics.**

| Variables | Frequency (N = 664) | Percent (100%) | Mean (SD) |
|---|---|---|---|
| **Age** | – | – | 28.82 (8.17) |
| **Gender** | | | |
| Male | 560 | 84.3 | |
| Female | 104 | 15.7 | |
| **Marital status** | | | |
| Single | 432 | 65.1 | |
| Married | 160 | 24.1 | |
| Cohabiting | 48 | 7.2 | |
| Divorced/Separated | 24 | 3.6 | |
| **Educational level** | | | |
| No formal education | 128 | 19.3 | |
| Basic education | 272 | 41.0 | |
| Secondary education | 232 | 34.9 | |
| Tertiary education | 32 | 4.8 | |
| **Native resident** | | | |
| Yes | 320 | 48.2 | |
| No | 344 | 51.8 | |
| **Living arrangements** | | | |
| Living alone | 336 | 50.6 | |
| Living with family | 264 | 39.8 | |
| Living with friends/colleagues | 64 | 9.6 | |
| **Average work duration** | – | – | 5.95 (5.30) |
| **Alcohol intake** | | | |
| Yes | 376 | 56.6 | |
| No | 288 | 43.4 | |
| **Self-rated health (SRH)** | | | |
| Excellent | 104 | 15.7 | |
| Very good | 96 | 14.5 | |
| Good | 200 | 30.1 | |
| Fair | 176 | 26.5 | |
| Poor | 88 | 13.3 | |

**Table 2. Means, standard deviations, and zero-order Pearson's correlations between the principal study continuous variables.**

| Variables | Mean (SD) | 1 | 2 | 3 | 4 |
|---|---|---|---|---|---|
| 1. WRS | 7.14(2.90) | 1 | | | |
| 2. SP | 5.25(2.32) | 0.371*** | 1 | | |
| 3. Anxiety | 8.22(4.82) | 0.499*** | 0.442*** | 1 | |
| 4. Depression | 10.36(5.69) | 0.522*** | 0.387*** | 0.586*** | 1 |

SD—standard deviation; WRS—work-related stress; SP—sleep problems.

***p < .001.

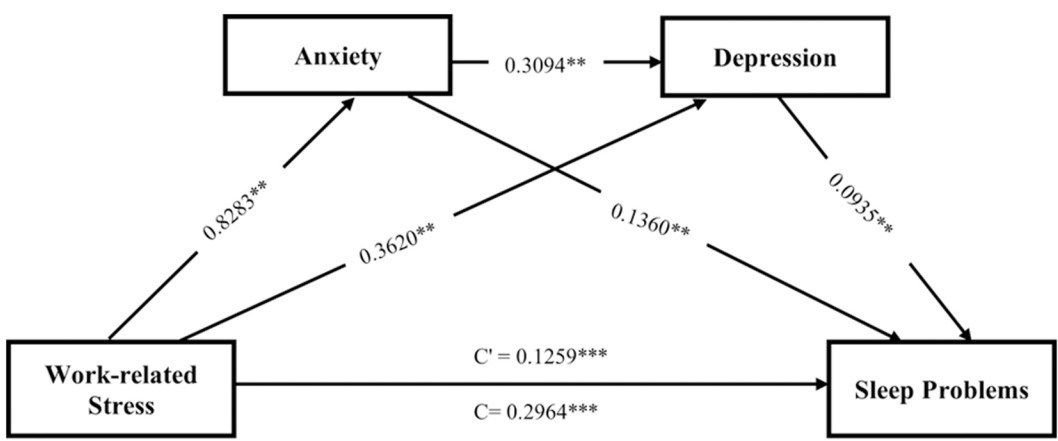

**Fig 3. A serial multiple mediation model of the association between work-related stress and sleep problems through anxiety and depression.**
Path unstandardized coefficients are reported. Each model was adjusted for age, sex, marital status, educational status, living arrangement, self-rated health, alcohol intake, and chronic conditions. **p<0.005, ***p<0.0001.

The analysis revealed that WRS had a significant positive association with increased levels of anxiety (B = .8283, 95% CI = .7186–.9380), depression (B = .3620, 95% CI = .2613–.3576), and SP (B = .1259, 95% CI = .0608–.1910). Anxiety, in turn, was significantly associated with heightened depression levels (B = .3094, 95% CI = .2613–.3576) and SP (B = .1360, 95% CI = .0947–.1773). Additionally, depression was found to be significantly linked to higher SP scores (B = .0935, 95% CI = .0347–.1524). The direct relationship between WRS and SP remained statistically significant even after the inclusion of the mediating variables, suggesting that anxiety and depression serve as partial mediators in the relationship between WRS and SP.

## Bootstrapping estimates

Table 3 presents the direct, indirect, and total effects of the relationship between WRS and SP among the gold miners in Ghana. The 95% CIs, generated using a bootstrap method with 5000 resamples, did not include zero for any of the effect estimates, indicating statistically significant indirect effects of WRS on SP through anxiety and depression, after adjusting for confounding variables. Three mediation pathways were identified: anxiety alone (B = 0.1127, BootSE = 0.0179, 95% CI = 0.0772 to 0.1471), which contributed 38.0% of the total effect; depression alone (B = 0.0339, BootSE = 0.0117,

**Table 3. The indirect effects of work-related stress on sleep problems through anxiety and depression as specific and chain mediators.**

| Path Model | B | BootSE | Boot 95%CI | % mediated |
|---|---|---|---|---|
| Direct effect | 0.1259 | 0.0332 | 0.2398 to 0.353 0 | 42.5% |
| *Indirect effects* | | | | |
| WRS -> anxiety -> SP | 0.1127 | 0. 0179 | 0.0772 0.1471 | 38.0% |
| WRS -> depression -> SP | 0. 0339 | 0. 0117 | 0.0124 0.0587 | 11.4% |
| WRS -> anxiety -> depression -> SP | 0. 0240 | 0. 0093 | 0.0081 0.0446 | 8.1% |
| *Total indirect effect* | 0.1705 | 0.0191 | 0.1338 0.2091 | 57.5% |
| *Total effect* | 0. 2964 | 0. 0288 | 0.2398 0.3530 | 100% |

*Note:* B–Unstandardized regression coefficients are reported; BootsSE–Bootstrapping standard error; WRS–work-related stress; SP–sleep problems. Each model was adjusted for age, sex, marital status, educational status, living arrangement, self-rated health, alcohol intake, and chronic conditions. The empirical 95% confidence interval does not overlap with zero.

95% CI = 0.0124 to 0.0587), accounting for 11.4%; and a serial pathway from anxiety to depression (B = 0.0240, BootSE = 0.0093, 95% CI = 0.0081 to 0.0446), which explained 8.1% of the total effect. Altogether, these indirect pathways constituted 57.5% of the overall effect of WRS on SP, while the remaining 42.5% was attributed to the direct effect.

## Discussion

This study examined the direct and indirect pathways linking WRS to SP among small-scale miners in Ghana, using a serial mediation model. The findings revealed that WRS was independently related with higher odds of SP, with anxiety and depression explaining this link both separately and sequentially; accounting for 38.0%, 11.4%, and 8.1% of the total effect, respectively. These results underscore the need for targeted interventions that address both psychological distress and sleep health in this occupational group. Although empirical studies specifically testing the WRS-SP association among miners are scarce, evidence from other occupational groups supports the notion that stressful and hazardous working conditions contribute to poor sleep health. For instance, Wu et al. [33] reported that among Chinese nurses, work stress affected sleep via circadian rhythm disruptions. Similarly, Yook [34] noted among 705 Korean male firefighters aged 40–50 years, occupational stress significantly correlated with cardiorespiratory fitness, arterial stiffness, and sleep quality. In the Netherlands, Van Laethem et al. [35] showed that work stress, rumination, and sleep quality were mutually reinforcing over time. Consistent with these studies, our results highlight the adverse implications of WRS for miners' sleep health and call for integrated psychosocial interventions in the mining sector.

The present analysis identified anxiety and depression as key psychological mechanisms mediating the relationship between WRS and SP. Anxiety accounted for approximately 38.0% of the indirect effect linking WRS to SP, suggesting that individuals experiencing heightened occupational stress are more likely to suffer sleep disturbances through increased anxiety. Consistent with previous research, chronic WRS has been associated with elevated levels of nervousness, physiological hyperarousal, and cognitive rumination, which are all hallmarks of anxiety episodes [36,37]. In hazardous work environments, such as mining and construction, continuous exposure to high demands without sufficient emotional or institutional support may trigger persistent anxiety episodes [38,39]. This heightened anxiety has been shown to impair sleep quality by increasing nighttime cortisol levels and activating the hypothalamic-pituitary-adrenal (HPA) axis [40,41]. Similarly, depression was found to mediate approximately 11.4% of the WRS-SP association. Thus, depressive episodes serve as a secondary but meaningful pathway linking occupational stress to impaired sleep health. Prior studies have established that prolonged exposure to work-stress contributes to psychological exhaustion and low mood, increasing vulnerability to depression [35]. Depression, in turn, has been consistently linked to sleep-related complaints such as early morning awakenings, circadian rhythm disturbances, and persistent feelings of unrest [42].

Our finding that WRS is indirectly associated with sleep disturbances through the serial mediation of anxiety and depression (accounting for 8.1% of the variance) appears to be novel, especially within the context of small-scale miners in Ghana; a population for which this specific relationship has not been previously explored. The analysis indicates that exposure to work stress may first trigger anxiety symptoms, which subsequently heighten depressive, ultimately impairing sleep quality. This pathway aligns with prior research showing that chronic occupational stress is often linked to increased anxiety and depression levels among workers [43,44]. Furthermore, in line with our results, earlier studies have demonstrated a strong connection between anxiety and depression, particularly in high-pressure work environments [45,46]. The co-occurrence of these psychological conditions tends to result in more pronounced sleep issues than when either condition is present alone, with reports indicating that up to 80% of affected individuals experience substantial sleep disturbances [47]. These findings highlight the importance of addressing both anxiety and depression as intervention targets, as their combined effect on sleep involves multiple mechanisms: anxiety contributes to initial sleep disruption through hyperarousal and elevated cortisol [48], while depression induces sleep problems via altered circadian rhythms and neurotransmitter imbalances [49].

Beyond anxiety and depression tested in this study, other potential mediating mechanisms could explain the relationship between WRS and SP among small-scale miners. Musculoskeletal disorders could be an important mediator [50,51]. The physically demanding nature of mining work, combined with work-related stress, leads to musculoskeletal pain and physical exhaustion [52], operating through both physical discomfort and elevated inflammatory markers that directly interfere with sleep architecture [53]. Again, substance use behaviors, particularly alcohol consumption and stimulant use, could mediate this relationship. Research among Australian miners found that those with high occupational stress were 2.5 times more likely to engage in hazardous alcohol consumption [54], with substances significantly impairing sleep quality by disrupting sleep-wake cycles [55]. The observed associations between the study variables may reflect an underlying pathway through which occupational stress is linked to psychological distress and impaired sleep health. However, the cross-sectional nature of this study limits our ability to infer causality or determine the temporal ordering of these variables. It is plausible, for example, that poor sleep may exacerbate depressive or anxiety symptoms, which in turn could increase perceived work stress. Again, individuals already experiencing psychological distress may be more likely to report sleep disturbances. It is well established that chronic sleep disturbances can impair emotional regulation and increase physiological reactivity to stressors, thereby intensifying stress perceptions [47]. Therefore, future longitudinal or experimental research is warranted to test and clarify the directionality of these associations, particularly within the context of small-scale mining.

Moreover, the broader Ghanaian context in which small-scale mining occurs may shape the observed associations. Small-scale miners in Ghana often operate in informal and unregulated settings characterized by hazardous working conditions, long hours, and limited access to healthcare and psychosocial support [56,57]. These conditions may intensify work-related stress and reduce opportunities for seeking help, thereby exacerbating both depression and sleep-related difficulties. Cultural expectations around masculinity and resilience, particularly within mining communities, may also discourage emotional disclosure and reinforce internalisation of distress [58,59], contributing to the underreporting of symptoms. Furthermore, the fear of job loss due to periodic government clampdowns on illegal mining activities adds an additional layer of insecurity, which may worsen psychological distress and impair sleep [60,61]. These contextual realities may partly explain the strength of associations observed in this study. As much, highlight the importance of developing interventions that are culturally appropriate, gender-sensitive, and tailored to the unique structural vulnerabilities of this population.

Furthermore, it is important to note that the associations observed in this study may be modified by individual-level characteristics such as gender, age, and years of mining experience. For instance, previous studies suggest that women often report higher levels of work-related stress and are more vulnerable to anxiety, depression, and sleep disturbances due to differences in hormonal regulation, caregiving burdens, and social expectations around emotional expression [62,63]. Similarly, younger or less experienced workers, including miners may have fewer coping resources or less job control, increasing their susceptibility to the negative effects of occupational stress [64]. In contrast, longer years of working or mining experience may lead to either psychological adaptation or cumulative stress exposure, depending on the individual and contextual factors [65]. Although these potential effect modifiers were not examined in the present study, they may influence the strength or direction of the pathways linking WRS to SP. Future research employing stratified analyses or moderated mediation models is warranted to assess the role of these variables and to identify subgroups that are particularly vulnerable to the mental health consequences of work-related stress.

Our findings have several implications for public health, mental health care, and policy intervention in LMICs. First, healthcare providers should incorporate mental health and work-related stress screenings into routine care for miners, offering tailored interventions to address anxiety and depression that are linked to sleep problems. Second, workplace-based programs aimed at reducing stress and promoting mental well-being should be implemented. These programs could include counseling services, stress management workshops, and cognitive behavioral therapy, which are specifically designed for the mining environment. Third, awareness programs targeting miners and employers about how

work-related stress leads to sleep problems through anxiety and depression can help foster a proactive approach to mental health care and reduce the stigma surrounding mental health issues. Finally, policy initiatives should focus on strengthening occupational health frameworks, including regular mental health screenings as part of health surveillance, and creating safer and more supportive work environments. Investing in organizational changes, such as reducing work-place stressors and offering support systems for miners, could help mitigate sleep disruptions and improve overall mental health outcomes.

Several strengths and limitations need to be acknowledged. First, our study is among the first to investigate the complex relationship between WRS and SP, with a specific focus on the serial mediating role of anxiety and depression among small-scale miners in Ghana. Importantly, we employed well-validated and widely used psychometric instruments to assess core constructs, including WRS, anxiety, depression, and SP, all with robust reliability and validity. Additionally, we controlled for a comprehensive set of potential confounders, such as demographic characteristics, health behaviors, and occupation-related variables, which enhances the internal validity and robustness of our findings. However, caution is warranted in interpreting the directionality of the observed associations due to the cross-sectional nature of the study. It remains unclear whether WRS causes SP through mental health pathways or if existing SP and mental health conditions exacerbate perceived stress levels. Future longitudinal studies are essential to clarify these temporal dynamics and disentangle the direction of causality between stress, anxiety, depression, and sleep disturbances. Also, reliance on retrospective self-report measures may introduce recall or social desirability biases, due to personal perception, stigma, or misunderstanding of items, potentially affecting the accuracy and strength of the observed associations. Moreover, although we accounted for several covariates, residual confounding from unmeasured factors, such as trauma history, physical illness, work hours, or shift work cannot be ruled out. To improve robustness and generalizability, future research should consider integrating objective assessments of work stress, sleep and mental health, along with more diverse and representative samples across mining communities.

## Conclusions

Using serial mediation modeling, this study provides new insights into the association between WRS and SP among small-scale miners. The findings reveal that WRS significantly impairs sleep quality, with anxiety and depression serving as key mediators. Notably, 57.5% of the WRS-SP relationship occurs through indirect pathways involving these psychological factors, both independently and sequentially. This highlights the complex interplay between occupational stress and sleep disturbances, emphasizing the need for comprehensive occupational health interventions. Effective strategies should integrate stress management programs, mental health support services, and sleep hygiene education tailored to mining workplaces. These measures could break the cycle of stress, psychological distress, and sleep problems experienced by many miners. Additionally, occupational health policies should expand beyond traditional safety concerns to prioritize mental health and sleep quality as critical components of worker well-being. We recommend future research to adopt longitudinal designs and examine diverse mining populations to confirm these findings and establish causality.

## Supporting information

**S1 File. Raw dataset used for statistical analysis in this study.**
(ZIP)

**S2 File. Inclusivity-in-global-research-questionnaire.**
(DOCX)

## Acknowledgments

We are grateful to all the study participants and all contributors for their support in completing this study.

## Author contributions

**Conceptualization:** Emmanuel Nyaaba, Lawrence Guodaar, Razak M. Gyasi.

**Data curation:** Emmanuel Nyaaba, Vanessa Francesca Epis, Lawrence Guodaar, Razak M. Gyasi.

**Formal analysis:** Emmanuel Nyaaba, Vanessa Francesca Epis, Lawrence Guodaar, Razak M. Gyasi.

**Investigation:** Emmanuel Nyaaba, Vanessa Francesca Epis, Lawrence Guodaar, Razak M. Gyasi.

**Methodology:** Emmanuel Nyaaba, Vanessa Francesca Epis, Lawrence Guodaar, Razak M. Gyasi.

**Project administration:** Emmanuel Nyaaba, Vanessa Francesca Epis, Lawrence Guodaar, Razak M. Gyasi.

**Resources:** Emmanuel Nyaaba, Vanessa Francesca Epis, Lawrence Guodaar, Razak M. Gyasi.

**Software:** Emmanuel Nyaaba, Vanessa Francesca Epis, Razak M. Gyasi.

**Supervision:** Lawrence Guodaar, Razak M. Gyasi.

**Validation:** Emmanuel Nyaaba, Vanessa Francesca Epis, Lawrence Guodaar, Razak M. Gyasi.

**Visualization:** Emmanuel Nyaaba, Vanessa Francesca Epis, Lawrence Guodaar, Razak M. Gyasi.

**Writing – original draft:** Emmanuel Nyaaba, Vanessa Francesca Epis, Lawrence Guodaar, Razak M. Gyasi.

**Writing – review & editing:** Emmanuel Nyaaba, Vanessa Francesca Epis, Lawrence Guodaar, Razak M. Gyasi.

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
