## [Decision Letter · Decision Letter 0]

5 Jun 2025

PONE-D-25-22418Work-related stress and sleep problems among small-scale miners in Ghana: The role of psychological factorsPLOS ONE

Dear Dr. Nyaaba,

Thank you for submitting your manuscript to PLOS ONE. After careful consideration, we feel that it has merit but does not fully meet PLOS ONE’s publication criteria as it currently stands. Therefore, we invite you to submit a revised version of the manuscript that addresses the points raised during the review process.

We look forward to receiving your revised manuscript.

Kind regards,

Mukhtiar Baig, Ph.D.

Academic Editor

PLOS ONE

3. In the online submission form, you indicated that [All data used in this study are available from the corresponding author upon reasonable request.].

Reviewers' comments:

Reviewer's Responses to Questions

**Comments to the Author**

1. Is the manuscript technically sound, and do the data support the conclusions?

Reviewer #1: Yes

Reviewer #2: Yes

Reviewer #3: Yes

2. Has the statistical analysis been performed appropriately and rigorously? 

Reviewer #1: Yes

Reviewer #2: Yes

Reviewer #3: Yes

3. Have the authors made all data underlying the findings in their manuscript fully available?

Reviewer #1: Yes

Reviewer #2: No

Reviewer #3: Yes

4. Is the manuscript presented in an intelligible fashion and written in standard English?

Reviewer #1: Yes

Reviewer #2: Yes

Reviewer #3: Yes

5. Review Comments to the Author

Reviewer #1: Thank you for giving me the opportunity to review this manuscript.

1) While the term "work-related stress" (WRS) is introduced, the definition is not clearly explained or supported by a more thorough background. It is better to describe more details or a broader literature context on how WRS is specifically measured or categorized in the studies referenced.

2) The introduction mentions that data on WRS and sleep problems (SP) in LMICs, particularly in Ghana, is scarce, but it does not explain why this gap is so significant. A clearer discussion of why it is particularly important to study WRS in LMICs (e.g., unique stressors, economic pressures) would strengthen the rationale.

3) Please describe any key elements such as the exposure (work-related stress), outcome (sleep problems), and mediators (anxiety, depression) should be more explicitly introduced at the beginning of the Methods section.

4) The total population size or sampling frame is not provided; thus, the representativeness of the sample is unclear. It is not stated whether an official registry or household listing was used to construct the sampling frame, which may raise concerns about selection bias. Please describe them.

5) Please describe any potential effect modifiers (e.g., gender, years of mining). Please describe how to assess their effects on clinical outcomes.

6) Please describe cultural and linguistic validity of standardized, validated instruments (PSS-4, PHQ-9, GAD-7, WHODAS 2.0) in the Ghanaian context.

7) Communities were purposively selected, possibly limiting generalizability. All data are self-reported; potential recall or reporting biases are not discussed. Residual confounding is not addressed, nor are any strategies (e.g., sensitivity analyses) mentioned to mitigate bias. Please describe those limitations in the discussions section.

8) Please describe the rationale for exceeding the minimum required sample (664 vs. 649).

9) Please describe any justification for household selection intervals (e.g., every third or fifth household).

10) Please describe how to handle missing values (e.g., listwise deletion, imputation).

11) The study draws conclusions about the causal pathway linking WRS → Anxiety → Depression → Sleep Problems, despite being cross-sectional.　The authors should explicitly state that the findings reflect associations rather than causality.

12) The discussion lacks acknowledgment of potential unmeasured confounders (e.g., trauma history, shift work, physical illness, work hours). Acknowledge limitations in controlling for all relevant variables that could influence both stress and sleep.

13) Several findings are repeated almost verbatim across multiple paragraphs (e.g., 80% sleep disturbance figure, sequential mediation mechanism). Streamline the discussion to avoid redundancy and enhance clarity.

14) Despite being conducted in Ghana, there is little reference to Ghana-specific working conditions, healthcare access, or cultural perceptions of stress and sleep. Please discuss how local socioeconomic or cultural factors may shape the findings or their applicability. It is important to discuss them to emphasize the novelty and clinical significance of this study.

15) Alternative interpretations (e.g., reverse causality where poor sleep leads to perceived stress) are not explored in depth. Please describe alternative hypotheses and how future studies could test them.

16) The discussion omits consideration of self-report bias, recall bias, or social desirability bias, which are highly relevant given the constructs measured. Please discuss how the use of self-reported psychometric tools may have influenced findings.

I think it is better to revise the manuscript.

Reviewer #2: Review comments

General comments

I commend the Authors for the good concept of this manuscript and the presentation also. I submit for a minor revision before the manuscript could be accepted for publication. Authors should take note of the following observations.

1. Journal guidelines. Authors should take note of Journal guidelines in presenting the abstract and also ensure that references and referencing style is uniform and also conform to Journal guidelines. (See lines 135 and 466).

2. Study design. The study should be referred to as a community based cross-sectional study. The ‘quantitative’ should stand for the method of data collection.

3. Sample size determination. Authors should justify the use of a default prevalence in determining the minimum sample size for the study. Authors are also commended for the ‘high’ sample size that was utilized for the study.

4. Sampling technique. What is the rationale for starting with purposive sampling and then continuing with probability sampling techniques? To this effect, the first stage as presented in the manuscript cannot be referred to as a stage since no probability sampling technique was applied. It is important to know whether the ‘availability of miners as stated in the inclusion criteria is normally distributed’ in the selected communities to warrant the application of probability sampling technique in selecting the respondents. What is the justification for using a sampling interval of five during field visits? Instead of using ‘randomly selected in line 162 it will be better to state that one miner was selected using a simple random sampling technique of balloting. Authors should explain why a cluster sampling technique was not applied in this study.

5. Authors are commended for the good description of the tools used in the study.

6. Taking into account the title of the study and its design, the inclusion of a conceptual framework will be of relevance.

7. The ‘Adjusted’ in line 40 should be deleted.

Reviewer #3: Abstract

The abstract may start with a sentence of two about the background or rationale for the study before outlining the objective.

Methods

Was incentive or any form of renumeration provided to the participants? This can be included in the methods section.

Discussion

Line 318 to 324. Is there any study linking occupational stress to impaired sleep outcomes specifically among miners? This can be included for effective comparison.

Minor Edits

Insert commas.

Line 42: The mean (SD) age was 28.8 (8.2) years, and 84.3% were males.

Line 83 This prolonged activation contributes to cumulative biological strain, which can ultimately result in serious health conditions [11].

Line 156 …the seven communities were purposively selected, as described above.

Also, check throughout the manuscript and edit where appropriate.

Change depressive to depression: Line 128: odds of SP and that this relationship will be serially mediated by anxiety and depressive.

Line 176 replace “a 8-week period” with “8 weeks”

Data collection was carried out over a 8-week period (from 16th February 2025 to 14th April.

6. PLOS authors have the option to publish the peer review history of their article (what does this mean? ). If published, this will include your full peer review and any attached files.

**Do you want your identity to be public for this peer review?** For information about this choice, including consent withdrawal, please see our Privacy Policy .

Reviewer #1: No

Reviewer #2: **Yes: ** EDMUND NDUDI OSSAI

Reviewer #3: No

---

## [Author Response · Author response to Decision Letter 1]

8 Jul 2025

REVIEWERS' COMMENTS AND RESPONSES

Dear Reviewers, we value your thorough and perceptive comments on our manuscript. To improve the manuscript's readability, clarity, and contribution, we have carefully considered every suggestion and made significant changes. Our answers to each of the points made are listed below;

REVIEWER #1:

Thank you for giving me the opportunity to review this manuscript.

1) While the term "work-related stress" (WRS) is introduced, the definition is not clearly explained or supported by a more thorough background. It is better to describe more details or a broader literature context on how WRS is specifically measured or categorized in the studies referenced.

Response

We thank the reviewer for this insightful comment. We have revised the introduction to provide a clearer and more detailed explanation of work-related stress (WRS), including how it is defined and commonly assessed in the literature. We have also updated the description of key studies to clarify how WRS was operationalized in each. We have indicated in the revised manuscript draft that:

“Defined by [9] as emotional, cognitive, behavioral, and physiological responses to negative aspects of work, WRS reflects a state of heightened arousal and distress. It often arises from persistent job demands such as excessive workload, role ambiguity, low job control, and employment insecurity [9]. Although short-term stress responses, such as fatigue, elevated heart rate, and negative affect, are common and typically harmless, they do not disrupt sleep when the body adequately recovers during off-job periods. However, sleep may be impaired when stress recovery is insufficient and physiological systems, such as the hypothalamic-pituitary-adrenocortical and sympathetic-adrenal-medullary axes, remain activated during rest periods [10]. This prolonged activation contributes to cumulative biological strain, which can ultimately result in serious health conditions [11]. WRS, including high-strain, low-control, or effort-reward imbalance situations without adequate recovery has been shown to predict poor sleep outcomes across various populations [12,13]. For instance, a mid-life Australian cohort study found that job stress, assessed via the Job Content Questionnaire (JCQ), independently predicted chronic sleep disturbances [12], while a systematic review of 38 studies confirmed a consistent negative association between WRS and sleep quality [14]. Among Chinese oilfield workers, occupational stress, which was assessed using the Occupational Stress Inventory-Revised (OSI-R) scale, capturing role overload, role ambiguity, and responsibility was positively associated with increased sleep [15]”

2) The introduction mentions that data on WRS and sleep problems (SP) in LMICs, particularly in Ghana, is scarce, but it does not explain why this gap is so significant. A clearer discussion of why it is particularly important to study WRS in LMICs (e.g., unique stressors, economic pressures) would strengthen the rationale.

Response:

We appreciate the reviewer’s thoughtful suggestion. We have revised the introduction to better articulate why the lack of research on WRS and sleep problems (SP) in LMICs, particularly in Ghana, represents a critical gap. We have indicated in the revised manuscript draft that:

“Crucially, while research has established clear links between occupational stress and sleep quality in formal work settings, the unique challenges faced by informal workers, especially in LMICs, remain understudied. This knowledge gap is crucial, given that LMICs often present unique work-related stressors, including informal employment arrangements, limited labor protections, low wages, high job insecurity, limited access to psychosocial support, and under-resourced health systems [6, 20,21], all of which may exacerbate the effects of WRS on physical and mental health. These contextual differences therefore suggest that findings from high-income countries may not fully capture the lived realities of workers in LMICs, where structural vulnerabilities heighten the risk of cumulative stress and poor sleep outcomes”

3) Please describe any key elements such as the exposure (work-related stress), outcome (sleep problems), and mediators (anxiety, depression) should be more explicitly introduced at the beginning of the Methods section.

Response:

We have revised the opening paragraph of the Methods section to explicitly introduce the key study variables.

4) The total population size or sampling frame is not provided; thus, the representativeness of the sample is unclear. It is not stated whether an official registry or household listing was used to construct the sampling frame, which may raise concerns about selection bias. Please describe them.

Response:

We are happy for this important observation. We have attempted to address the query in the revised manuscript draft. Specifically, we have clarified that prior to data collection, a reconnaissance survey was conducted to identify and enumerate households with active miners. Working in collaboration with local residents and district authorities, we developed a comprehensive household listing that served as the sampling frame. This listing included 1,970 houses.

5) Please describe any potential effect modifiers (e.g., gender, years of mining). Please describe how to assess their effects on clinical outcomes.

Response:

We do appreciate this beautiful idea shared by the reviewer to consider some effect modifiers in our analysis. While we agree with the reviewer based on our understanding on the role of effect medication in proximate relationships, we would humbly like to indicate that the focus on the moderating effect of potential variables falls outside the scope of this paper. We have considered this as the main focus of our other paper which is currently being considered in another destination. We hope that the consolidation of these papers will definitely proffer complementarity to enrich our understanding just as you indicated. However, we have provided a general description of the background characteristics of our study, including gender, age and mining experienced in the discussion section of the revised draft.

6) Please describe cultural and linguistic validity of standardized, validated instruments (PSS-4, PHQ-9, GAD-7, WHODAS 2.0) in the Ghanaian context.

Response:

We have attempted to address this issue in the methods section detailing the cultural and linguistic validity of the standardized instruments used (PSS-4, PHQ-9, GAD-7, and WHODAS 2.0). We have indicated in the revised draft that:

“All measurement tools used in this study (PSS-4, PHQ-9, GAD-7, and WHODAS 2.0) have been previously applied in studies conducted in Ghana [5] or among similar populations in LMICs [16,17], showing acceptable psychometric properties. Where necessary, minor contextual adaptations were made to enhance cultural appropriateness. Instruments were administered in English or translated into Twi using forward and backward translation procedures by bilingual experts. This ensured conceptual and linguistic equivalence. Additionally, the tools were pre-tested among a small group of 25 artisanal miners to ensure clarity, relevance, and cultural sensitivity before full-scale data collection”

7) Communities were purposively selected, possibly limiting generalizability. All data are self-reported; potential recall or reporting biases are not discussed. Residual confounding is not addressed, nor are any strategies (e.g., sensitivity analyses) mentioned to mitigate bias. Please describe those limitations in the discussions section.

Response:

We appreciate these thoughtful comments. We have revised the “Strengths and Limitations” section to explicitly acknowledge the potential limitations arising from the study design, self-reported data, and unmeasured confounders.

8) Please describe the rationale for exceeding the minimum required sample (664 vs. 649).

Response:

In the revised manuscript, we have clarified the rationale for exceeding the minimum required sample size. Although the scientifically estimated minimum sample size was 649 (accounting for a 35% nonresponse rate), we extended data collection to 664 participants to further strengthen the statistical power of the study and account for any unforeseen data quality issues such as incomplete responses or outliers so that we don’t miss our minimum sample target.

9) Please describe any justification for household selection intervals (e.g., every third or fifth household).

Response:

We have provided a clear justification for the use of varying selection intervals. This stratified interval approach was primarily based on community size and housing density to ensure balanced and proportional representation across all selected communities. Additionally, we acknowledge that the term “household” was not the most appropriate and have corrected it to “house(s)” throughout the methods section for clarity. Thank you for the insights.

10) Please describe how to handle missing values (e.g., listwise deletion, imputation).

Response:

Thank you for your insightful comment. We have attempted to address this query by specifying our approach to handling missing data in the statistical analysis section. The section now reads as:

“All analyses were conducted using complete case analysis (listwise deletion), as missing data were minimal (<5%) and determined to be missing completely at random (MCAR) based on Little’s MCAR test”

11) The study draws conclusions about the causal pathway linking WRS → Anxiety → Depression → Sleep Problems, despite being cross-sectional.　The authors should explicitly state that the findings reflect associations rather than causality.

Response:

Thank you for this valuable observation. We agree that the cross-sectional design of the study does not permit causal inference. In response, we have revised the discussion section to explicitly acknowledge that our findings shows associations rather than causality, and have demonstrated potential bidirectionality. The revised draft now reads as follows:

“The observed associations between the study variables may reflect an underlying pathway through which occupational stress is linked to psychological distress and impaired sleep health. However, the cross-sectional nature of this study limits our ability to infer causality or determine the temporal ordering of these variables. It is plausible, for example, that poor sleep may exacerbate depressive or anxiety symptoms, which in turn could increase perceived work stress. Again, individuals already experiencing psychological distress may be more likely to report sleep disturbances. It is well established that chronic sleep disturbances can impair emotional regulation and increase physiological reactivity to stressors, thereby intensifying stress perceptions [47]. Therefore, future longitudinal or experimental research is warranted to test and clarify the directionality of these associations, particularly within the context of small-scale mining.”

12) The discussion lacks acknowledgment of potential unmeasured confounders (e.g., trauma history, shift work, physical illness, work hours). Acknowledge limitations in controlling for all relevant variables that could influence both stress and sleep.

Response:

We appreciate these thoughtful comments. We have revised the “Strengths and Limitations” section to explicitly acknowledge the potential limitations arising from the self-reported data, and unmeasured confounders such as trauma history, shift work, and physical illness.

“Also, reliance on retrospective self-report measures may introduce recall or social desirability biases, due to personal perception, stigma, or misunderstanding of items, potentially affecting the accuracy and strength of the observed associations. Moreover, although we accounted for several covariates, residual confounding from unmeasured factors, such as trauma history, physical illness, work hours, or shift work cannot be ruled out. Furthermore, no sensitivity analyses were conducted to assess the robustness of our findings to potential sources of bias. To improve robustness and generalizability, future research should consider integrating objective assessments of work stress, sleep and mental health, along with more diverse and representative samples across mining communities”

13) Several findings are repeated almost verbatim across multiple paragraphs (e.g., 80% sleep disturbance figure, sequential mediation mechanism). Streamline the discussion to avoid redundancy and enhance clarity.

Response:

We appreciate the reviewer’s observation regarding repetition in the discussion section. In response, we have thoroughly revised the discussion to eliminate redundancy and improve clarity.

14) Despite being conducted in Ghana, there is little reference to Ghana-specific working conditions, healthcare access, or cultural perceptions of stress and sleep. Please discuss how local socioeconomic or cultural factors may shape the findings or their applicability. It is important to discuss them to emphasize the novelty and clinical significance of this study.

Response:

Thank you for this insightful suggestion. In response, we have integrated a new paragraph within the Discussion section that situates our findings within the broader socio-cultural and structural context of small-scale mining in Ghana. The new paragraph reads as follows:

“Moreover, the broader Ghanaian context in which small-scale mining occurs may shape the observed associations. Small-scale miners in Ghana often operate in informal and unregulated settings characterized by hazardous working conditions, long hours, and limited access to healthcare and psychosocial support [56,57]. These conditions may intensify work-related stress and reduce opportunities for seeking help, thereby exacerbating both depression and sleep-related difficulties. Cultural expectations around masculinity and resilience, particularly within mining communities, may also discourage emotional disclosure and reinforce internalization of distress [58,59], contributing to the underreporting of symptoms [75]. Furthermore, the fear of job loss due to periodic government clampdowns on illegal mining activities adds an additional layer of insecurity, which may worsen psychological distress and impair sleep [60,61]. These contextual realities may partly explain the strength of associations observed in this study. As much, highlight the importance of developing interventions that are culturally appropriate, gender-sensitive, and tailored to the unique structural vulnerabilities of this population”

15) Alternative interpretations (e.g., reverse causality where poor sleep leads to perceived stress) are not explored in depth. Please describe alternative hypotheses and how future studies could test them.

Response:

Thank you for this insight. We have now expanded the discussion and have attempted to address this query by including alternative interpretations of the observed associations. The revised draft now reads as follows:

“The observed associations between the study variables may reflect an underlying pathway through which occupational stress is linked to psychological distress and impaired sleep health. However, the cross-sectional nature of this study limits our ability to infer causality or determine the temporal ordering of these variables. It is plausible, for example, that poor sleep may exacerbate depressive or anxiety symptoms, which in turn could increase perceived work stress. Again, individuals already experiencing psychological distress may be more likely to report sleep disturbances. It is well established that chronic sleep disturbances can impair emotional regulation and increase physiological reactivity to stressors, thereby intensifying stress perceptions [47]. Therefore, future longitudinal or experimental research is warranted to test and clarify the directionality of these associations, particularly within the context of small-scale mining”

16) The discussion omits consideration of self-report bias, recall bias, or social desirability bias, which are highly relevant given the constructs measured. Please discuss how the

---

## [Editor Report · Decision Letter 1]

22 Jul 2025

Work-related stress and sleep problems among small-scale miners in Ghana: The role of psychological factors

PONE-D-25-22418R1

Dear Dr. Nyaaba,

We’re pleased to inform you that your manuscript has been judged scientifically suitable for publication and will be formally accepted for publication once it meets all outstanding technical requirements.

Kind regards,

Mukhtiar Baig, Ph.D.

Academic Editor

PLOS ONE

---

## [Editor Report · Acceptance letter]

PONE-D-25-22418R1

PLOS ONE

Dear Dr. Nyaaba,

I'm pleased to inform you that your manuscript has been deemed suitable for publication in PLOS ONE. Congratulations! Your manuscript is now being handed over to our production team.

Kind regards,

on behalf of

Professor Mukhtiar Baig

Academic Editor

PLOS ONE